# Effects of Adding Neutralized Red Mud on the Hydration Properties of Cement Paste

**DOI:** 10.3390/ma13184107

**Published:** 2020-09-16

**Authors:** Sukpyo Kang, Hyeju Kang, Byoungky Lee

**Affiliations:** 1Department of Architecture, Woosuk University, Jincheon 27841, Korea; ksp0404@woosuk.ac.kr; 2Department of Construction Engineering, Woosuk University, Jincheon 27841, Korea; 3COCHEMS Co., Ltd. Industrial Tools Circulating Center, 160, Daehwa-ro, Daedeok-gu, Daejeon 34368, Korea; fluolbk@naver.com

**Keywords:** red mud, liquefied red mud, neutralized red mud, cement paste, heat of hydration, compressive strength, X-ray diffraction, recycling

## Abstract

Red mud is a highly alkaline waste by-product of the aluminum industry. Although recycling of red mud is being actively researched, a feasible technological solution has not been found yet. In this study, we propose that neutralization of red mud alkalinity could assist in its use as a construction material. Neutralized red mud (LRM + S; pH 6–8) was prepared by adding sulfuric acid to liquefied red mud (LRM; pH 10–12). After adding LRM and LRM + S to cement paste, the heat of hydration, compressive strength, and hydration products were examined. The experiments revealed that the calorific value of the cement paste with LRM was low, and its peak was delayed, when compared with that of plain cement paste (referred to as Plain), whereas the calorific value of the cement paste with LRM + S was similar to that of Plain. At the age of 28 d, the compressive strength of the cement paste with 10% LRM + S was 99% whereas that with 20% LRM was only 55% of the strength of Plain. Thus, our results help to resolve the issue of strength degradation of cementitious materials observed upon the addition of red mud and enable its reuse as a construction material.

## 1. Introduction

The development of industrial and mass-production processes has resulted in an increased generation of waste, and the disposal of such waste has been recognized as a serious environmental issue. The development of recycling technologies to convert large quantities of waste into natural materials may provide opportunities to mitigate the waste-management problem [1]. Red mud is a reddish-brown sludge by-product generated through the Bayer process in the aluminum industry. When producing 1.0 ton of alumina, 1.0–1.5 tons of red mud is generated as a by-product [2,3]. The chemical components of red mud are Al_2_O_3_ (17%–20%), Fe_2_O_3_ (48%–54%), SiO_2_ (4%–6%), TiO_2_ (3%–4%), Na_2_O (3%–5%), and CaO (1%–2%) [3]. The disposal of red mud is becoming a major problem for industries, as it can lead to soil and water pollution owing to its high alkalinity [4].

Every year, 660 million tons of red mud is estimated to be generated worldwide due to aluminum and its ancillary industries [5,6]. While the processing and recycling of red mud has been a subject of active research, no feasible technology has yet been developed [7]. Meanwhile, 280,000 tons of red mud is currently being annually generated in South Korea, and the recycling rate is less than 10%, thereby requiring the development of a technology that can recycle large quantities of red mud. In South Korea, red mud is designated as a general waste according to the Enforcement Regulations of Waste Management Act [8]. In the construction industry, it can be reused as a cement admixture material in cement, concrete, ready-mixed concrete, and ceramic products.

While the construction industry can help recycle red mud in large quantities; however, an analysis of the strength characteristics of red mud is important before it can be utilized as a construction material. Previous studies have reported that adding red mud to cement paste and mortar increases their compressive strengths [6,9]. This is thought to be due to the slight pozzolanic reaction and the matrix filling effect of fine red mud particles. On the other hand, Ribeiro et al. mentioned that replacing 35% and 50% of cement with red mud reduced compressive strength by approximately 23% and 64%, respectively, compared to the standard sample. They reported that this is due to the limited hydration characteristics of red mud [1].

Moreover, owing to its high Fe_2_O_3_ content, red mud can also be utilized as a red pigment. Korean regulations specify that red mud can be recycled as a pigment and a coloring agent. Accordingly, some of the red mud generated in South Korea is recycled as a coloring agent. In the recycling process, red mud is dried and crushed, whereby its moisture content, of 30–50 wt% at the time of discharge, is reduced to less than 10 wt% [10]. Consequently, a large amount of additional energy is required for this process, thereby increasing the overall recycling cost. Previously, with the intention of reducing the recycling cost of red mud, the authors of this study studied the production of liquefied red mud (LRM) by adding water to red mud with 36% moisture content such that it may be recycled as a cement concrete material. However, the results of the study revealed that the addition of LRM to cement concrete lowered the strength of the concrete [11]. The decreased compressive strength with increase in alkali content was in agreement with previous findings from literature, and it was attributed to the porous microstructure and lower strength of the alkali-containing C–S–H gel of hardened concrete developed in the high-alkali condition [12].

Therefore, we wanted to examine whether the neutralization of LRM could solve the reduction of concrete strength, using LRM derived from prior research results. In this study, in an attempt to improve the strength degradation observed earlier, we neutralized LRM to reduce its high pH (from 10–12 to 6–7). For experimental validation, we compared added LRM and neutralized LRM (LRM + S) to cement paste and examined the heat of hydration, compressive strength, and hydration products of the samples using X-ray diffraction (XRD). We expect the results of our study to contribute to an increase in the recycling of red mud as a construction material.

## 2. Experimental Program

### 2.1. Materials

In this study, LRM and neutralized red mud (LRM + S) were prepared using red mud sludge (KC, Korea). LRM was prepared by mixing red mud sludge, water, a thickener, and an antifoamer in the ratio of 1:0.2:0.0036:0.0014 based on the mass of the red mud sludge with a moisture content of approximately 36 wt%. The red mud sludge was first mixed with water for approximately 3 min using a homomixer, as illustrated in Figure 1. The thickener and antifoamer were then added and mixed for 2 min to improve the storage stability [11].

LRM + S was prepared by adding sulfuric acid (95% purity) to LRM with a pH of 11.5 such that the pH could be maintained at 6–8 after a day.

Table 1 details the physical properties of LRM and LRM + S. The pH values of LRM and LRM + S were 11.5 and 6.7, respectively. Table 2 and Figure 2 present the results of X-ray fluorescence (XRF) and XRD analyses of LRM and LRM + S, respectively. As shown in Table 2, the SO_3_ content of LRM + S was approximately 4.19% higher than that of LRM owing to the addition of sulfuric acid. Figure 2 shows that the main compounds of LRM are quartz, calcite, boehmite, and hematite [13]. These compounds were also observed in LRM + S. For LRM + S, characteristic peaks were observed at 2θ = 25.5 and 51.1 (estimated to be representative of gypsum and sodium sulfate, respectively) owing to the addition of sulfuric acid.

Ordinary portland cement (OPC, KS L 5201) was used in this study, and its physical and chemical properties are presented in Table 3. The compressive strength of cement was 29.5 MPa at the age of 3 d, 43.8 MPa at the age of 7 d, and 58.9 MPa at the age of 28 d.

### 2.2. Experimental Plan

Table 4 details the mixing compositions of the cement pastes used in this study. For the plain cement paste (referred to as Plain) mixture, only cement was used, and the water-to-cement ratio was set at 0.3. Such a minimal water-to-cement ratio was selected to help generate a large amount of hydration products for examining the strength and hydration characteristics. For the red mud mixtures, the addition of LRM and LRM + S were 10 and 20% by weight of cement. We used 10% and 20% LRM because the strength sharply decreased when LRM content in the cement was increased beyond 20%. This was also confirmed by our preliminary experiments. To prepare the cement pastes, mixing was performed using a mortar mixer for 4 min.

### 2.3. Methods

A multichannel microcalorimeter was used to measure the flow of heat of hydration. When LRM and LRM + S were used, they were added to water and then mixed with cement [14]. The flow of heat of hydration was measured from the moment when the water was first mixed with cement up until 72 h had elapsed.

Samples for the compressive-strength measurement were fabricated with dimensions of 40 mm × 40 mm × 160 mm. They were removed from the mold after a 24-h curing process at a temperature of 20 ± 2 °C and relative humidity of 50% and then cured under these same temperature and humidity conditions until they reached the desired age for obtaining the compressive-strength measurements. Compressive strength was measured under the same conditions for three samples of each mixture, and the average compressive strength was obtained at ages of 1, 3, 7, and 28 d. The compressive strength measurement process was in accordance with the standards of ASTM C 349 (Universal Testing Machine).

To examine the hydration products, the samples were collected at various ages (1 h and 1, 3, 7, and 28 d) and immersed in ethyl alcohol for 24 h to stop hydration. They were then dried in an oven at 80 °C for 24 h. The dried samples were crushed and sifted through a 200-mesh sieve for XRD analysis [15,16]. The XRD analysis mainly compared the availability of portlandite among the hydration products to analyze the hydration characteristics of cement using LRM before and after neutralization.

## 3. Results and Discussion

### 3.1. Heat of Hydration

Heat of hydration of cement can be classified into heat of hydration over time and accumulated heat of hydration. In general, the heat of hydration over time is used to indirectly predict the setting time of the cement. Similarly, the accumulated heat of hydration is used to predict the initial compressive strength of the cement paste [17].

Figure 3a shows the evolution of heat of hydration over time. In general, the heat of hydration of cement paste has two peaks. The first peak is generally related to the formation of ettringite (AFt) [18,19]. The second peak is related to the hydration of C_3_S and C_2_S, as well as the formation of C-S-H and portlandite [15]. In this study, the first and second peaks occurred in 0.1–0.3 h and 15–54 h, respectively.

Figure 3b shows the first peaks. The first peak for Plain occurred at approximately 0.1 h. The first peaks of the cement paste samples with LRM and LRM + S occurred at approximately 0.15 and 0.1 h, respectively. Compared to that of Plain, the first peaks of the LRM and LRM + S samples were similar and delayed, respectively. This indicates that the hydration rate related to the formation of ettringite can be improved by neutralizing the LRM with sulfuric acid.

Figure 3c shows the second peaks. The second peak of Plain occurred at approximately 15 h. The second peaks of the paste samples with LRM occurred at 24 and 54 h for LRM10 and LRM20, respectively. They were delayed by 1.6–3.6 times in comparison with that of the second peak of Plain. Conversely, the second peaks of the paste samples with LRM + S were similar to that of Plain regardless of the amount of additive. This indicates that the hydration of C_3_S and C_2_S, as well as the formation of C–S–H and portlandite, are similar to those processes that occur in Plain when LRM is neutralized with sulfuric acid and added to the cement paste.

Figure 4 shows the accumulated heat of hydration over time. The accumulated heat of hydration of Plain over 72 h was 9.8 J/g. When LRM was added, the accumulated heat of hydration was found to be 9.5 J/g for LRM-10 and 6.4 J/g for LRM-20. In the case of LRM + S, the accumulated heat of hydration was 10.2 J/g for LRM + S-10 and 9.5 J/g for LRM + S-20. This indicates that an initial strength higher than that of LRM can be achieved when LRM is neutralized with sulfuric acid and added to cement paste, regardless of the added amount.

### 3.2. Compressive Strength

Figure 5 presents the compressive strength measurement results. The compressive strength of Plain at the age of 28 d was 61 MPa. The compressive strengths of LRM10 and LRM20 at the same age were 38 and 34 MPa, respectively, which were 38% and 44% lower in comparison to that of Plain.

Conversely, the compressive strengths of LRM + S10 and LRM + S20 at the same age were 60 and 53 MPa, respectively, which were 59% and 56% higher in comparison with that of the LRM samples.

Figure 6 plots the evolution of compressive strength ratios of the LRM and LRM + S samples in comparison with the compressive strength of Plain. As the amount of LRM added increased, the strength ratio decreased. Before 7 d, the difference in strength ratio between LRM10 and LRM20 increased as the age decreased. This appears to have occurred because LRM had a negative effect on the initial hydration of cement, resulting in a negative effect on the strength ratio, which increased as the amount of additive increased. In particular, the strength degradation of LRM20 at ages of 1 and 3 d was noticeable, which agrees with the result that the accumulated heat of hydration of LRM20 was relatively low, as shown in Figure 4. The negative effects of LRM on the initial hydration also appear to have affected the compressive strength at the age of 28 d. In previous studies, the use of red mud was limited to 10% or less for the construction industry because red mud does not undergo pozzolanic reactions [1,20,21].

In the case of LRM + S, the strength ratio decreased as the amount of additive increased. As opposed to the LRM samples, the differences in strength ratios between LRM + S10 and LRM + S20 at all ages were similar. Moreover, the negative effect on the initial hydration of the cement decreased in comparison with LRM and the strength ratio slightly increased at the age of 28 d. The fine particles of red mud appeared to have had a positive effect on the compressive strength as they filled the matrix [1,22]. The compressive strength of LRM + S10 at the age of 28 d was similar to that of Plain, indicating that the content of red mud can be increased further than the previously presented value (by an amount of less than 10%) by neutralizing the red mud [1,21].

### 3.3. Microstructure

Figure 7 presents the XRD results obtained for different ages. Portlandite is the main hydration product of cement, and calcium silicate is the unhydrated substance [23].

As shown in Figure 7a, portlandite, the main hydration product, was observed at 2θ = 18.1°, 34.1°, and 47.2° at 1 h for Plain. In general, portlandite is continuously observed in cement paste after watering. However, in the case of the cement paste with added LRM and LRM + S, the main peak of portlandite was not observed at 2θ = 18.1°. This appears to be due to the presence of Na-based compounds in red mud; this agrees with the results of a previous study wherein portlandite was not initially observed when Na-based compounds were added to cement paste [24]. In addition, owing to the first peak appearing 1 h earlier in the heat of hydration result (Figure 3a), ettringite was observed in 1 h even in the XRD result.

Figure 7b shows that the cement paste with LRM did not exhibit the Ca(OH)_2_ peak at 2θ = 18° and 1 d, but the peak was observed at 2θ = 18° when LRM + S was added. As LRM + S was prepared by neutralizing LRM with sulfuric acid, Na_2_SO_4_ was observed in the XRD pattern analysis. According to a previous study, the addition of a small amount of sulfate to cement paste delays the hydration of the cement, whereas the addition of a large amount of sulfate accelerates it [25]. This agrees with our results, thereby confirming that the cement paste with LRM + S exhibited a compressive strength closer to that of Plain in comparison with the cement paste with LRM at the age of 1 d, and that the cement paste with LRM did not exhibit a similar strength.

In Figure 7c,d, the cement paste with LRM exhibited the Ca(OH)_2_ peak at 2θ = 18° at 3 and 7 d. This agrees with the compressive-strength results, thereby confirming that the strength of LRM-20 sharply increased from 3 d onward. Additionally, the hydration of C_3_S and C_2_S, as well as the generation of C–S–H and portlandite, occurred while the second peak was observed in the heat of hydration results.

Figure 7e shows that calcite (CaCO_3_), portlandite (Ca(OH)_2_), and calcium silicate hydrate (C-S-H), which serve to develop strength, were observed in Plain, LRM, and LRM + S at the age of 28 d.

When red mud was added to cement concrete and XRD analysis was conducted, no new phase was observed even when LRM and LRM + S were added to the cement paste, unlike in a previous study [24].

## 4. Conclusions

In this study, neutralized red mud was prepared by neutralizing liquefied red mud (LRM) with sulfuric acid to remediate the issue of strength degradation observed in cementitious materials when mixed with highly alkaline LRM. The heat of hydration, compressive strength, and hydration products of LRM and LRM + S cement pastes were compared using X-ray diffraction (XRD), and the following conclusions were drawn:(1)The heat of hydration measurement results indicated that the maximum heat of hydration peak exhibited a low calorific value and was delayed for the cement paste with LRM in comparison with Plain. Conversely, the heat of hydration peak of the cement paste containing LRM + S was found to be similar to that of Plain.(2)The compressive strength at the age of 28 d for the cement paste with LRM was found to be as low as 55% of the compressive strength of Plain, whereas the cement paste with LRM + S showed a strength ratio as high as 99%.(3)In the XRD analysis, the cement paste with 20% LRM + S, unlike in the case of LRM sample, showed a Ca(OH)_2_ peak after only 1 h, similar to the case of Plain. In addition, no new products due to the addition of red mud were observed.(4)As the compressive strength of LRM + S10 at 28 d was similar to that of Plain, it can be assumed that (under the same conditions as in this study) up to 10% neutralized red mud can be added to cement pastes without affecting its strength.

Therefore, the results of our study show that red mud could be considered as a viable construction material if it is suitably processed by reducing its alkalinity. In addition, we are testing other acids, and the results will be reported in a future work. We also note that it is necessary to further clarify the results reported in this paper through analysis of the hydration products.

## Figures and Tables

**Figure 1 materials-13-04107-f001:**
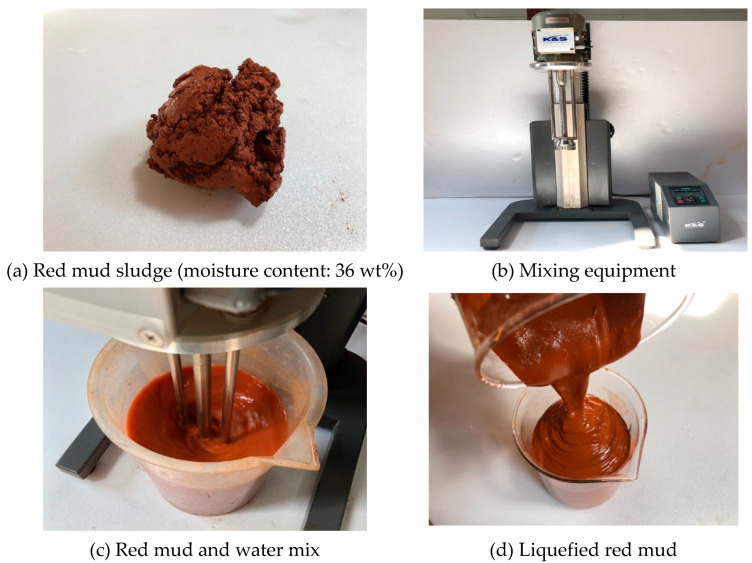
Manufacturing process of liquefied red mud.

**Figure 2 materials-13-04107-f002:**
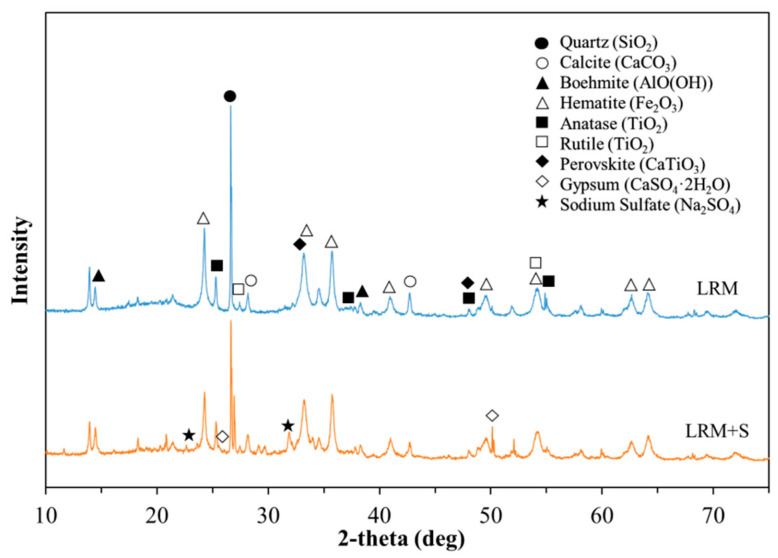
XRD patterns of LRM and LRM + S.

**Figure 3 materials-13-04107-f003:**
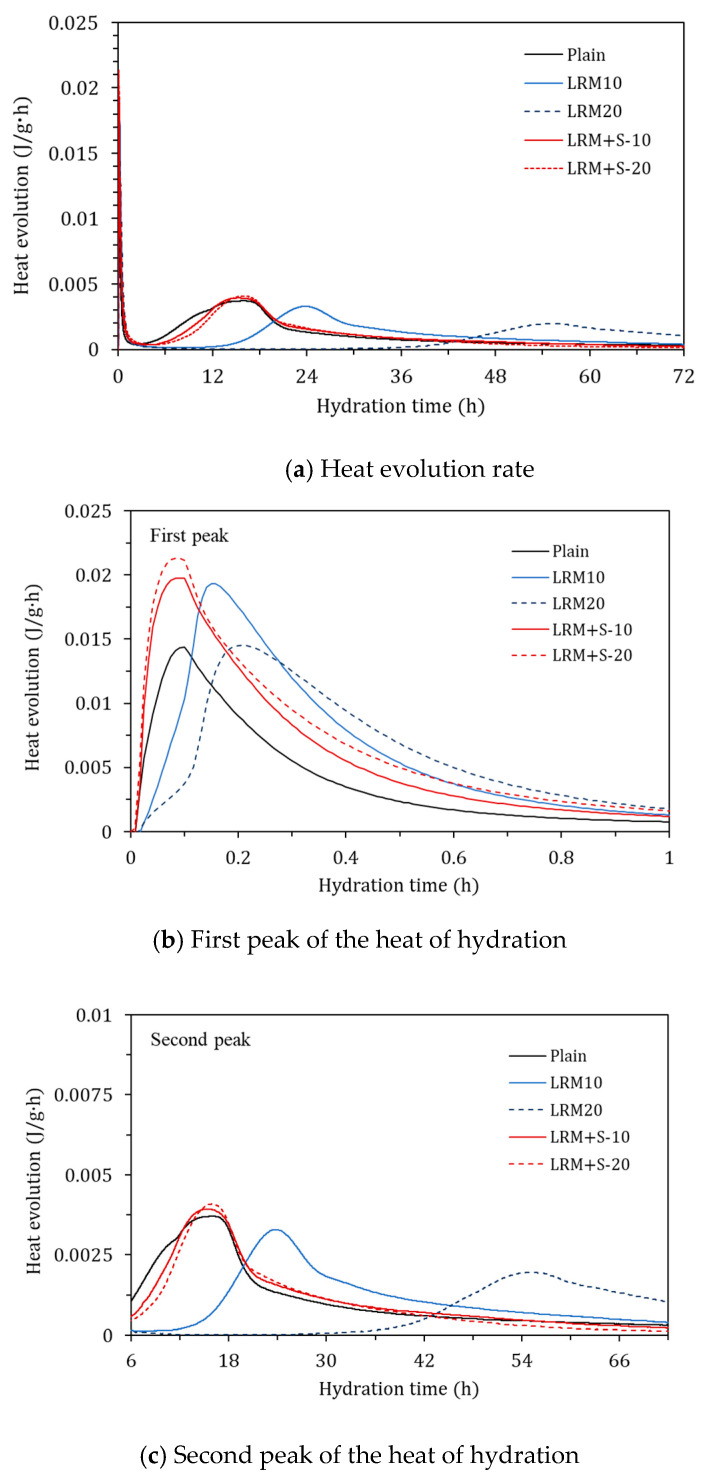
Heat evolution rates of the various cement paste samples.

**Figure 4 materials-13-04107-f004:**
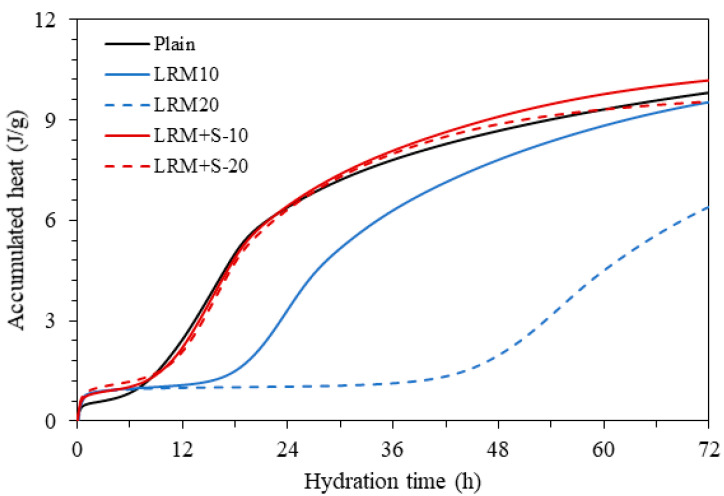
Accumulation of heat for the various cement paste samples.

**Figure 5 materials-13-04107-f005:**
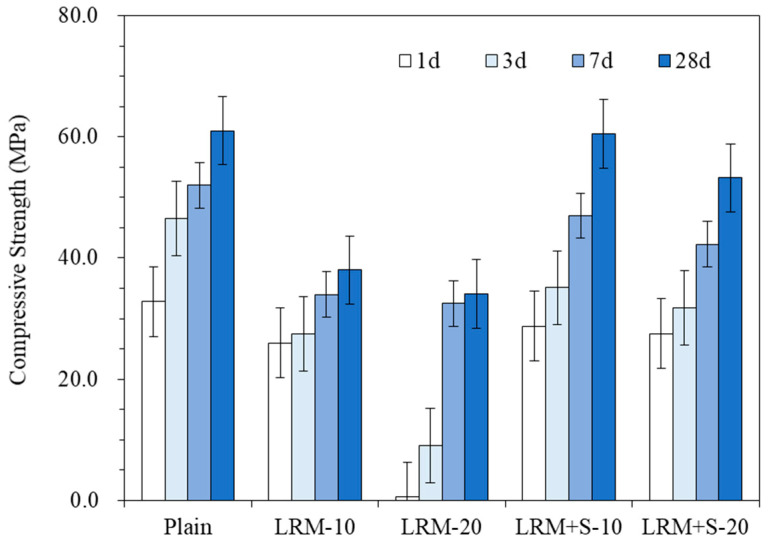
Effects of LRM and LRM + S on the compressive strength of the cement paste.

**Figure 6 materials-13-04107-f006:**
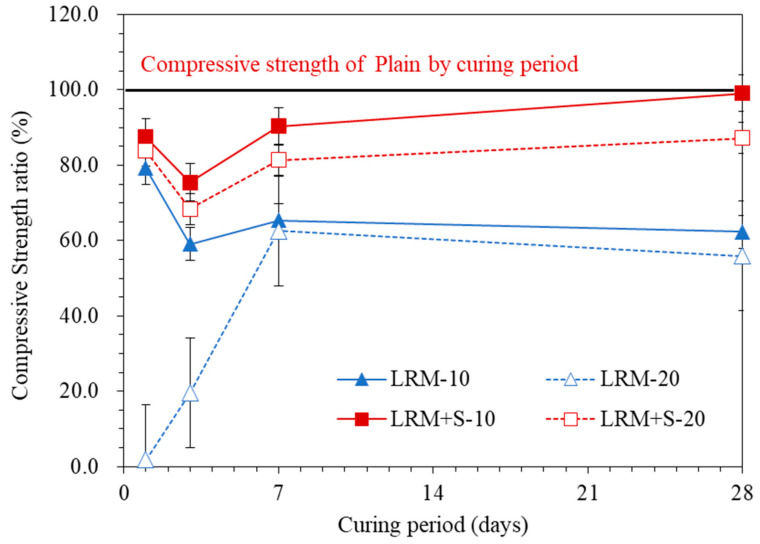
Effects of LRM and LRM + S on the compressive strength ratio of the cement paste.

**Figure 7 materials-13-04107-f007:**
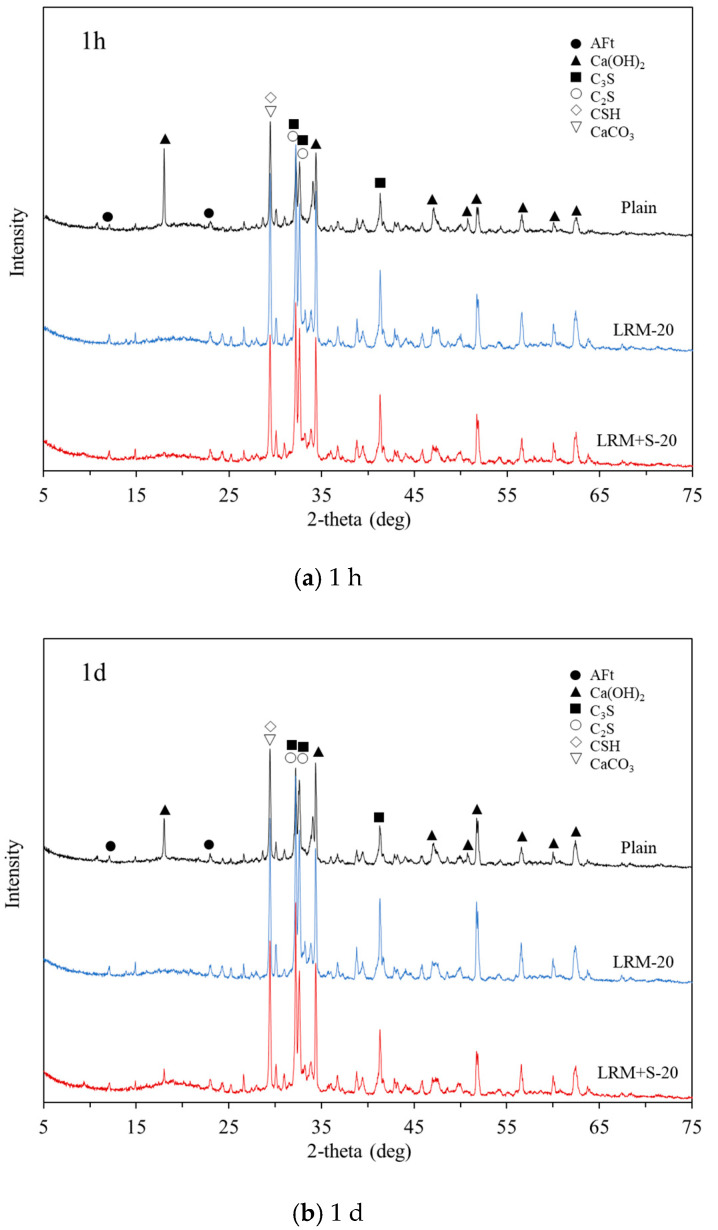
XRD spectrum results of the mixtures with red mud at varying ages.

**Table 1 materials-13-04107-t001:** Physical properties of red mud.

Type of Red Mud	Moisture Content (%)	pH	Density (g/cm^3^)	Specific Surface Area (cm^2^/g)	Average Particle Diameter (μm)	Viscosity (cP)
LRM *	49.5	11.5	1.5	2353	2.75	42,550
LRM + S **	49.2	6.7	1.54	2353	2.75	45,250

* LRM: Liquefied red mud; ** LRM + S: Liquefied red mud + sulfuric acid.

**Table 2 materials-13-04107-t002:** Chemical composition of red mud.

Type of Red Mud	SiO_2_	A1_2_O_3_	Fe_2_O_3_	MgO	Na_2_O	CaO	TiO_2_	SO_3_
LRM	17.6	25.6	30.4	0.21	13.2	1.83	6.27	0.29
LRM + S	17	25.4	29.2	0.21	10.7	1.84	5.99	4.48

**Table 3 materials-13-04107-t003:** Physical properties and chemical composition of OPC.

Type	Blaine (cm^2^/g)	Setting Time		Density (g/cm^3^)	Chemical Composition (%)						
		Initial (min)	Final (min)		SiO_2_	Al_2_O_3_	Fe_2_O_3_	CaO	MgO	SO_3_	lg. Loss
OPC *	3300	200	330	3.15	21.7	5.7	3.2	63.1	2.8	2.2	2.44

* OPC: Ordinary portland cement.

**Table 4 materials-13-04107-t004:** Mixture design.

Mix ID	Cement (wt%)	Water (wt%)	Extra Discount Red Mud (wt%)	
			LRM	LRM + S
Plain	100	30	-	-
LRM-10			10	-
LRM-20			20	-
LRM + S-10			-	10
LRM + S-20			-	20

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
