# Peer review of "Effects of Adding Neutralized Red Mud on the Hydration Properties of Cement Paste"

_materials, 2020, doi:10.3390/ma13184107_

Round 1

Reviewer 1 Report

The paper presents a study on the effect of adding neutralized red mud on the hydration properties of cement paste. In general, the paper is well written; however, the reviewer has some below comments.

  1. Introduction:

In fact that there is a huge study on the utilization of red mud in the concrete field. The authors have addressed the importance to utilize of red mud; however, the authors did not indicate (mention) the limitation of previous studies as well as did not show clearly the purpose of this study. The introduction part must be revised carefully to address the novelty to this research.

  1. Materials and methods

The authors should delete all the name of company in (…).

Why did the authors decide to used 10 and 20% of LRM? In addition, why the authors used sulfuric acid instead of other acids.

Line 108-109: “….humidity of 50 % and then cured under the same conditions until they reached the
109 desired age for obtaining the compressive-strength measurements”, what is the same curing condition here, please add the detail?

To examine the hydration products, the authors used XRD analysis; however, according to the knowledge of the reviewer, XRD can detect minerals but sometimes it is quite sensitive to have a good result of Portlandite peaks (maybe lowe intensity). In addition, why the author did not use thermal analysis, this technique can detect Portlandite easily.

The author dried the samples using the oven at 80oC, however, this temperature can destroy the structure of the sample, may be related to CSH and ettringite. In general, the drying temperature is lower than 50 oC is acceptable. Besides, in general, we use the vacuum pump to dry the specimen before crushing the sample, this technique can eliminate the destruction of the sample structure.

  1. Results and discussion.

Lines 130-132 and 136-138: Please add the reason why adding LRM and or adding sulfuric acid, it caused the delay of hydration? In addition, why the second peak of heat evolution of  LRM20 (Fig. 3c) or accumulated heat (Fig. 4) was the lowest?

Figs. 2 and 7, please remove the number in the vertical axis. Because the author has combined two or more  XRD patterns in one figure, thus, the value of intensity (counts) has no meaning anymore.

Fig. 5. Please check the standard deviation (error bar) of compressive strength, please make it consistently.

Title of 3.3 may not be correct Because of 3.1. Hydration heat, 3.2. Compressive strength, thus 3.3 should be related to minerals or microstructure, not XRD (XRD is just of a test method to examine somethings….)

Line 18: check again English of this sentence because of the repetition of “and”.

Line 48-52: please add the citation of these statements.

Reviewer 2 Report

This paper illustrates the results of a close analysis of neutralized red mud (LRM+S) at a pH of 6–8 was prepared by adding sulfuric acid to liquefied red mud (LRM) at a pH of 10–12. After adding LRM and LRM+S to the cement paste, the hydration heat, compressive strength, and hydration products were examined. The observed accumulated  hydration heat revealed that the calorific value of the cement paste with LRM was low and its and peak was delayed when compared with that of plain cement paste (referred to as Plain), whereas the calorific value of the cement paste with LRM+S was similar to that of Plain. More specifically about the strength degradation of cementitious materials was improved by adding neutralized red mud prepared by adjusting the pH of highly alkaline LRM with sulfuric acid.

In my opinion it is a very interesting study and I recommend that the paper be published subject to materials journals.

Some Minor Comments and Suggestions for Authors:

  • Minor corrections of grammar and spelling. The authors are required to check the whole manuscript with a grammar specialist as it has several grammatical errors.

  • I also feel as if some of the results are being held back and a more detailed analysis of the competing materials should be given so that the paper can be more useful to a wider audience. In my opinion the mineralogical investigation it is very important for the cement paste and the possibility of cohesion between the cement paste and the surface of aggregate particles.

  • The authors have done a great job on the literature review. Please add more literature with regards to the works that have been published in the field of the influence of mineralogical composition on the quality of cement. Please check the follow topic :

  • Theofani Tzevelekou, Paraskevi Lampropoulou, Panagiota P. Giannakopoulou, Aikaterini Rogkala, Petros Koutsovitis, Nikolaos Koukouzas, Petros Petrounias. Valorization of slags produced by smelting of metallurgical dusts and lateritic ore fines in manufacturing of slag cements. Applied Sciences 2020. mdpi.

Generally in my opinion in this manuscript needed extended analysis in the introduction part about the petrographic/mineralogical analysis of the cement paste or materials on the final compressive strength of paste. Also in the discussion part needed more information about the potential behavior of aggregate particles and this cement paste in the concrete specimens.

  • Please provide the exact specification of the cement paste material and strengths.
  • Please provide a more detailed reasoning behind the structure behavior. The details should include the rigid numbers or percentages.
  • What is the reason of choosing those W/C ratios? Please extended explain it in the manuscript.
  • If you like add error bars to the figures. Please do this for the rest of the figures. It is not so important but it is more feature of academic writing.
  • The presentation and discussion of results is clear and concise.
  • Conclusion needs to be more concise. Please use fewer sentences containing percentages and illustrate the main conclusions in the manuscript. The article describes a consistent experimental program. Based on the experimental results, some relevant conclusions are presented.

In general I consider the article to be of very good quality and should be accepted for publication after minor revisions.

Reviewer 3 Report

This article is a partial follow-up to a previously published article on the topic of liquefied red mud and I have the following comments:

  • line 85 - please specify cement designation
  • table 2 - please specify final setting time in the same units as innitial setting time
  • line 97 - why was the w/c ratio 0.3 chosen?
  • point 4 in the conclusion is vaguely defined by how much the amount of LRM + S can be increased - please rework
  • citations 9 and 10 are identical
  • the citation format does not match the template for the journal

Reviewer 4 Report

This research aims at improving the strength of cementitious materials with adding LRM and neutralized red mud (LRM+S). Authors have employed a thorough experimental work and they were successful in analysing the obtained results. However, some points need to be addressed:

  1. The submitted manuscript needs proofreading. Many mistakes were submitted e.g. (In the Abstract, line 18: "...with LRM was low and its and peak was delayed").
  2. The introduction part needs to be extended and enhanced. Authors have mentioned that red mud is being recycled and used as admixture material. Authors should mention other admixture materials and technologies that are currently used for concrete enhancement and improvement e.g. flax/wool twine, nano-Graphite (nG), and sodium acetate.
  3. In section 2.2 (experimental plan), what is the reason behind using w/c ratio of 0.30 ?
  4. In section 2.2 (experimental plan),authors mentioned that the addition of LRM and LRM + S was 10% or 20% by the weight of cement. Was this addition a replacement of cement or just addition ? and if it was addition, did authors consider it in the mix design and w/c ratio calculations?  Please clarify in the text. Also, it will be more appropriate to say the addition of LRM and LRM + S was 10% and 20% by the weight of cement.

  5. In section 2.3 (Methods), mention the total number of samples used in research and the number of samples used for each test.
  6. The title of section 3.1 (hydration heat) is not scientifically accurate. Please change it to (heat of hydration).
  7. In section 3.1 (hydration heat), authors mentioned that the hydration rate related to the formation of ettringite can be improved by neutralizing the LRM with sulfuric acid. It is well-known that ettringite is formed mainly from sulfate (calcium aluminium sulfate), and authors have used sulfuric acid to neutralize LRM. It is expected that ettringite will be increased not reduced. Furthermore, the increase of ettringite will lead to accelerate the deterioration rate of concrete. Please, clarify.

Authors need to address the mentioned comments before accepting the manuscript.

Round 2

Reviewer 1 Report

The authors have tried to revise the manuscript however, there are still some issues that need to be solved.

Point 1:

The authors tried to address the limitation of previous works, but the authors still have not yet pointed out the novelty of this research, I can say the introduction part is still needed to improve.

In addition, a small point about tracking the change in the revised part, it is not a professional way to use the tracking in the revised part. Instead of that, the authors should highlight the revised part, which makes it easier for the reviewer as well as editors.

Point 2:

The authors have answered in the cover letter, however, the author should also revise in the manuscript (the reason why the author used 10 and 20% of LRM).

The authors mentioned that “ Tests are currently underway with the other acids, and the test results will be reported in the future”, why the author did not mention at the end of the conclusion as a recommendation for future work?

Point 6: Related to XRD test

The response of the author is not fair enough, it is needed to explain in a scientific way; in addition, the author must revise it and add the limitation and recommendation for future works at the end of the conclusion.

Point 7: Related to 80 oC

The response of the author is not fair enough, because it is needed to answer in a reasonable way, not just only to cite the reference.

Round 3

Reviewer 1 Report

Thank you very much for addressed all comments of reviewers.

The quality fo the paper has been improved significantly. It can now be accepted for publication.

Thank you.